# Multimodal Treatment of Metastatic Rectal Cancer in a Young Patient: Case Report and Literature Review

**DOI:** 10.3390/medicina60050696

**Published:** 2024-04-24

**Authors:** Ionuț Popescu, Ana-Maria Dudău, Simona Dima, Vlad Herlea, Vlad M. Croitoru, Ioana Mihaela Dinu, Monica Miron, Ioana Lupescu, Irina M. Croitoru-Cazacu, Radu Dumitru, Adina Emilia Croitoru

**Affiliations:** 1Faculty of Medicine, Titu Maiorescu University, 040441 Bucharest, Romania; r.ionut.popescu@gmail.com (I.P.); vlad.m.croitoru@gmail.com (V.M.C.); 2Medical Oncology Department, Fundeni Clinical Institute, 022328 Bucharest, Romania; mihagerard@yahoo.com (I.M.D.); monica.miron2015@gmail.com (M.M.); irina.cazacu89@gmail.com (I.M.C.-C.); adina.croitoru09@yahoo.com (A.E.C.); 3Faculty of Medicine, University of Medicine and Pharmacy Carol Davila, 020021 Bucharest, Romania; dima.simona@gmail.com (S.D.); herlea2002@yahoo.com (V.H.); ilupescu@gmail.com (I.L.); radu.dumitru@gmail.com (R.D.); 4Center of Excellence in Translational Medicine, Fundeni Clinical Institute, 022328 Bucharest, Romania; 5Pathology Department, Fundeni Clinical Institute, 022328 Bucharest, Romania; 6Radiology Department, Fundeni Clinical Institute, 022328 Bucharest, Romania

**Keywords:** metastatic rectal cancer, early onset colorectal cancer, liver metastases, conversion therapy, liquid biopsy, bone metastases

## Abstract

Metastatic colorectal cancer requires a multidisciplinary and individualized approach. Herein, we reported the case of a young woman diagnosed with metastatic rectal cancer who received an individualized multimodal treatment strategy that resulted in a remarkable survival. There were several particular aspects of this case, such as the early onset of the disease, the successful use of conversion therapy, the application of liquid biopsy to guide treatment, and the specific nature of the bone metastasis. To offer more insights for navigating such challenges in patients with metastatic colorectal cancer, we have conducted a literature review to find more data related to the particularities of this case. The incidence of early onset colorectal cancer is on the rise. Data suggests that it differs from older-onset colorectal cancer in terms of its pathological, epidemiological, anatomical, metabolic, and biological characteristics. Conversion therapy and surgical intervention provide an opportunity for cure and improve outcomes in metastatic colorectal cancer. It is important to approach each case individually, as every patient with limited liver disease should be considered as a candidate for secondary resection. Moreover, liquid biopsy has an important role in the individualized management of metastatic colorectal cancer patients, as it offers additional information for treatment decisions.

## 1. Introduction

Colorectal cancer is the third most common cancer worldwide, with 1.1 million new cases per year, making it the second leading cause of cancer death [1]. Approximately 20% of newly diagnosed patients have synchronous liver metastases, while 50% develop them over the course of their illness [2,3]. Apart from the liver, the lung, peritoneum, and lymph nodes are also frequent metastasis locations.

Before initiating any treatment, a clinical or biological assessment of metastatic colorectal cancer (mCRC) must be conducted through appropriate radiological imaging and histological examination of either the primary tumor or its metastases [4]. Evaluating mismatch repair (MMR) status and identifying mutations in KRAS, NRAS exons 2, 3, and 4, along with BRAF mutations, are essential steps recommended for all metastatic patients at diagnosis to guide the choice of initial therapy. RAS mutations serve as negative predictors for the efficacy of anti-epidermal growth factor receptor (EGFR) monoclonal antibodies, while the BRAF V600E mutation is a significant adverse prognostic factor, making it essential to evaluate both BRAF and RAS mutations together for accurate prognostic determination [5,6,7]. Assessing for deficient mismatch repair (dMMR) or microsatellite instability (MSI) plays a key role in genetic counseling, such as detecting Lynch syndrome, and is vital in the preliminary molecular analysis for choosing patients suitable for immune checkpoint inhibitor treatment [8,9,10].

When approaching a patient with mCRC, a pivotal aspect of the case management is determining whether the individual is a candidate for possible surgical resection. In certain instances, removal of the primary tumor may be required due to symptoms such as obstruction or bleeding. Although retrospective studies indicate potential advantages of primary tumor resection, even in the absence of these critical symptoms [11], randomized trials specifically investigating this issue have not shown a survival benefit for patients with inoperable coexisting metastases [12,13]; therefore, this strategy is not typically recommended.

The R0 surgical removal of liver metastases is considered a potentially curative approach. Oncological factors, such as the onset of metastatic disease (synchronous versus metachronous), the presence of extrahepatic disease, and the response to neoadjuvant treatment, provide prognostic information that can predict a longer progression-free survival or a higher likelihood of cure [4]. Additionally, surgical or technical criteria, which are not constrained by the number, size, or presence of tumors in both lobes of the liver, indicate that liver resection is feasible if over 30% of the liver mass remains intact post-operation, or if the liver-to-body mass ratio exceeds 0.5 [14,15,16].

Although very few randomized, controlled studies have been conducted, the observation that patients who are initially considered unresectable become resectable after responding to systemic therapy has led to the introduction of conversion therapy into clinical practice. Despite the fact that patients who require conversion therapy have a worse prognosis than those who can be directly operated on, receiving the surgery improves their overall survival [16]. Therapeutic decisions are predominantly determined by the RAS mutation status and the location of the primary tumor. Therefore, for patients with RAS wild-type and left-sided tumors, the preferred approach is the combination of anti-EGFR antibodies with a cytotoxic platinum-based doublet regimen [17,18]. On the other hand, for patients with RAS-mutant disease, particularly those with right-sided tumors, a cytotoxic triplet regimen plus Bevacizumab has shown to elicit the strongest overall response [19,20,21].

In instances of unresectable metastatic disease, the initiation of systemic therapy is essential. Historically, the foundation of such treatment has been 5-fluorouracil (5-FU) [22]. Enhancing this efficacy of this drug by combining it with oxaliplatin and/or irinotecan has shown to improve response rates and survival outcomes [23,24]. Recent advancements in the field have introduced targeted therapies, underscoring the importance of identifying predictive biomarkers (such as RAS and BRAF mutations, and dMMR/MSI-H status) before initiating treatment. Two distinct anti-EGFR agents have shown efficacy in metastatic disease, either as standalone treatments or in combination with chemotherapy. Cetuximab, a chimeric monoclonal antibody, has proven to be effective across various treatment lines, similarly to Panitumumab [25,26]. In the first-line setting, Bevacizumab, a selective inhibitor of vascular endothelial growth factor A, is the sole antiangiogenic agent that has demonstrated improved results when used alongside chemotherapy [27]. Another cornerstone moment was the coming-of-age for immunotherapy and successfully identifying a predictive biomarker. Currently, Pembrolizumab represents a standard of care in naive of treatment patients who have a deficient mis-match repair mechanism (dMMR) or a high microsatellite instability (MSI-h) [9]. Collectively, these agents represent the cornerstone of first- and second-line treatments in the management of mCRC.

Treatment options for third-line therapy and beyond can be adjusted based on several criteria, including the patient’s performance status, the toxicities from previous treatments, and the history of prior therapies. Apart from the well-established use of Trifluridine-tipiracil, recently with improved responses when associated with Bevacizumab [28], small-scale studies have investigated the strategy of rechallenging patients with anti-EGFR antibodies, if they previously experienced significant progression-free survival during the initial treatment. The CRICKET phase II trial utilized liquid biopsy sampling to evaluate RAS and BRAF mutations, aiming to forecast the effectiveness of rechallenging patients with cetuximab. This approach successfully identified patients who were still RAS wild-type and achieved response during treatment [29].

Liquid biopsy has emerged in recent years as a powerful and attractive tool that involves collecting and analyzing cancer-derived materials, for example circulating tumor cells (CTCs) and circulating tumor DNA (ctDNA), from peripheral blood or other body fluids like ascites, urine or even pleural effusion. This approach focuses on the genomic or proteomic assessment of these materials to aid in the diagnosis, monitoring, and potentially the treatment of cancer [30]. The advancements in technologies for detecting genetic abnormalities have significantly emphasized the importance of liquid biopsy in the realm of precision oncology. It offers a more effective way to address the challenges posed by tumor heterogeneity compared to traditional tissue biopsies, enabling more frequent and less invasive testing. This method allows for the continuous, real-time monitoring of a patient’s tumor at the molecular level, providing insights into tumor burden and genetic mutations throughout the disease’s progression. Such capabilities are invaluable for tailoring and modifying subsequent treatment strategies, underscoring the wide-ranging clinical potential of liquid biopsy.

Undoubtedly, treating metastatic colorectal patients is a challenging and everchanging domain, which should involve key specialists in the treatment of digestive cancers (such as oncologists, gastroenterologists, pathologists, surgeons, radiotherapists, and radiologists). The clinical outcomes were highly improved when patients were treated in high-volume centers and regularly submitted to multidisciplinary reviews [31].

The aim of this article is to present the evolution and multimodal treatment of a complex case of metastatic rectal cancer in a young patient. We have also conducted a literature review to find relevant data related to the particularities of this case.

## 2. Case Report

In 2016, a 38-year-old female with no significant medical history was diagnosed in our clinic with an obstructing tumor in the medium rectum (6–10 cm from the external anal orifice), that could not be surpassed with the colonoscope. The main symptom was constipation. She had ECOG 1 performance status. The level of CA 19-9 was increased (479 UI/mL) (Figure 1).

An abdominopelvic MRI revealed multiple liver metastases between 3 and 30 mm in diameter, replacing nearly 50% of the liver (Figure 2 and Figure 3). The case was discussed in a multidisciplinary team, and the proposed strategy was surgical intervention for the primary tumor, followed by systemic therapy. A low anterior rectal resection with an end-terminal mechanical colorectal anastomosis was performed. The pathological exam described a tubulo-papillary adenocarcinoma, G1, pT3, pN1c (10 lymph nodes (LNs) positive out of 13 LNs removed, with tumor deposits in the perirectal tissue) and a Ki67 index of 68%. The immunohistochemistry (IHC) report results were: all-RAS wild type, BRAF V600 negative, MSS, and HER2/neu negative. After surgery, systemic treatment with a platinum-based chemotherapy doublet combined with anti-EGFR monoclonal antibodies (cetuximab) was initiated. The side effects were a grade 3 acneiform rash and a grade 2 sensory neuropathy. After three months, a CT scan showed a partial response in the liver metastases. The case was discussed in a multidisciplinary team. Despite the partial response, the liver metastases were still unresectable, and the decision was to continue the treatment, with de-escalation to modified DeGramont and cetuximab. In September 2017, another partial response was obtained (Figure 4), and the patient was referred to the surgical department. Eight liver metastases were removed from segments 2, 3, 4A, 4B, 5, and 8 (three tumors). The pathological examination of all tumor fragments confirmed metastatic colorectal adenocarcinoma. Both the pathological report and liquid biopsy confirmed that the tumor was all-RAS wild type.

In September 2018, increased levels of the tumoral biomarkers were detected, and the abdominopelvic MRI and PET-CT confirmed progressive liver disease. Chemotherapy with modified DeGramont and cetuximab was reinitiated, and a second metastasectomy was possible due to another partial response. After surgery, the patient received systemic treatment with capecitabine and oxaliplatin (CAPOX). After three months of CAPOX, imaging evaluation showed progressive disease with liver and pulmonary metastases. Between March 2019 and January 2020, second-line therapy with 5-FU, leucovorin, irinotecan (FOLFIRI) and Aflibercept was initiated with stable disease for 10 months. Unfortunately, progression was documented in January 2020 during a thoracic CT scan. Another liquid biopsy was taken, and again revealed no KRAS, NRAS, or BRAF mutations. Given the previous positive success with anti-EGFR therapy, we considered a rechallenge with cetuximab monotherapy to be appropriate. After five months, the liver metastases increased in size and number, and a new metastatic site was discovered in the left talar bone (Figure 5). The tumor measured 3 cm in its largest dimension. A bone biopsy confirmed a metastasis of an intestinal adenocarcinoma. The talar tumor was treated with external beam radiation (IMRT 40 Gy, split into 10 fractions). Systemic therapy with 5-FU, leucovorin and oxaliplatin (FOLFOX) was reinitiated, but the patient developed hypersensitivity to the oxaliplatin infusion, even at morefold dilutions, so we switched to an irinotecan-based regimen. Blood tests revealed a rising trend for the CA 19-9 biomarker in May 2021, and the MRI revealed progression of the liver tumors, which was confirmed at the multidisciplinary team meeting. Fourth-line treatment with trifluridin-tipiracilum was initiated with a two-month progression-free interval. The patient then developed speech and movement difficulties, and a brain MRI revealed a 35 mm diameter right frontal lobe mass with perilesional edema, which was surgically removed. 

It is important to emphasize that IHC evaluations were conducted for both bone and cerebral metastases. In both instances, the findings indicated an intestinal adenocarcinoma as the primary site. This conclusion was supported by the following marker profile: CK 7 was negative, CK 20 was positive, CDX2 showed diffuse positivity in tumor cells, and Ki67 levels were between 80 and 90%. Conversely, a comprehensive molecular assessment was not undertaken, specifically omitting the reevaluation of RAS mutations and microsatellite stability, with the exception of the liquid biopsy performed before the patient’s rechallenge with the anti-EGFR therapy.

We decided to collect another liquid biopsy, this time for FoundationONE next generation sequencing due to the long-term evolution, numerous therapy lines, and good performance status. The blood mutation burden was 18 muts/MB, which was equivocal for a treatment option such as immunotherapy. Besides the TMB, the NGS panel also exposed several mutations like KRAS G12D and G13D, APC H1329fs*2, DNMT3A Q573*, MRE11A E460*, SDHA K547*, TP53 R248Q.

Unfortunately, the patient’s clinical status deteriorated. She was referred to palliative care and died within two months. Throughout the progression of the illness and the various changes to her treatment regimen, the patient was able to sustain an acceptable quality of life. Notably, she endured some grade 3 allergic reactions, including an acne-like rash and sensory neuropathy, following the initial administration of an anti-EGFR agent in conjunction with platinum-based chemotherapy. To address these adverse effects and improve her clinical condition, we initiated targeted antibiotic therapy, which led to the resolution of the reactions shortly after their onset. Remarkably, the patient continued to exhibit a stable appetite and performance status, with no significant weight fluctuations, and was able to maintain her employment activities up until the identification of brain metastases, thereby preserving her quality of life for the majority of her disease trajectory.

## 3. Discussions

Herein, we reported the case of a young woman diagnosed with metastatic rectal cancer who received an individualized multimodal treatment strategy that resulted in a remarkable survival. There are several particular aspects of this case, such as the disease’s early onset, the use of conversion therapy, the application of liquid biopsy to guide treatment, and the specific nature of the bone metastasis. To offer more insights for navigating such challenges in patients with colorectal cancer, we have also conducted a literature review to find more data related to the particularities of this case

**Early-onset colorectal cancer (EOCRC)** is a world-wide phenomenon affecting individuals under 50 years of age, with an increased incidence of nearly 30% over the last two decades, especially in high-income countries like the United States, Australia and Canada [32]. Since 1994, EOCRC incidence rates have been escalating at an approximate rate of 2% annually. The frequency of EOCRC is inversely related to age [33], and is particularly concerning given the general decrease in both the overall incidence and mortality rates of colorectal cancer [34]. Additionally, there is a consensus that EOCRC is distinct from late-onset colorectal cancer (LOCRC, in patients older than 50 years), in terms of its pathological, epidemiological, anatomical, metabolic, and biological characteristics [35].

In terms of localization, EOCRC is primarily found in the rectum, followed closely by the distal colon, with over 70% of these cancers located on the left side of the colon at the time of diagnosis [36]. This observation provides insights into their characteristics, origins, and treatment approaches. Specifically, cancers located on the left side tend to be smaller, exhibit lower rates of recurrence, have longer periods of survival without the disease compared to those on the right side and most importantly, adhere to different treatment protocols [37]. Overall, men have a 30% higher incidence of colorectal cancer compared to women. This gender disparity is even more underlined in cases of rectal over colon cancer.

Approximately 30% of EOCRCs can be attributed to family history and hereditary factors. The overall mutational load in EOCRC is estimated at 16%, with Lynch Syndrome mutations representing half of this percentage. The remainder consists of various other mutations, including those in the adenomatous polyposis coli (APC) gene, monoallelic and biallelic mutations in MutYH, and mutations in BRCA1/BRCA2 [38]. Lynch Syndrome arises from a hereditary mutation in one of the mismatch repair genes (MLH1, MSH2, MSH6, PMS2, or EPCAM), leading to deficient MMR which mostly correlates with high microsatellite instability. Current protocols advise the universal testing of all newly identified colorectal cancer cases for MMR/MSI status to identify those affected by it and to offer proper treatment [9,39]. Another two very important genetic mechanisms affected in CRC are chromosomal instability (CIN) and CpG island methylator phenotype (CIMP) pathways. The chromosomal instability (CIN) pathway is marked by a buildup of mutations in tumor-suppressor genes and oncogenes, such as APC, KRAS, and TP53, contributing to 85% of sporadic cases [40]. The CIMP pathway and the BRAF V600E mutation are considered key molecular features of the serrated pathway, typically linked with lesions in the proximal colon [41]. Even so, approximately 70% of EOCRCs are sporadic. Deep learning algorithms that incorporate surrogate biomarkers, environmental exposures, genetic profiles, microbiome compositions, inflammation levels, and other factors contributing to EOCRC will enhance our comprehension of the disease. Risk models incorporating these elements have already been created for LOCRC groups and hereditary conditions like Lynch syndrome. Although these models represent significant progress, their precision, measured by sensitivity and specificity, still falls short of the desired standards.

Adjustable risk factors, organized by their relative risk from highest to lowest, encompass a diet typical of Western countries, smoking status (comparing those who currently smoke to never-smokers), intake of red and processed meats, and being overweight or obese [42,43]. On the other hand, regular physical activity has evidence of lowering the risk of colorectal cancer [44], while a sedentary lifestyle is linked with a heightened risk of EOCRC [45]. Findings from both an observational study and a prospective cohort study indicate that obesity during adolescence is linked to a higher occurrence and mortality rate [46,47]. However, there is also data that suggest no significant risk from obesity, so further prospective studies are warranted to clearly identify it as a risk factor in EORTC. Another important observation would be the impact of gut microbiome. Consuming a Western-style diet and maintaining an obese state may cause imbalances in gut microbiota and ongoing inflammation in the intestines, factors that could promote the development of CRC. This type of cancer is linked to a decrease in gut bacteria responsible for producing short-chain fatty acids, essential for maintaining the immune balance within the intestines [48]. In an effort to enhance the understanding of adjustable risk factors, European researchers have launched the Colorectal Cancer Pooling Project, aiming to consolidate data from over 25 prospective cohort studies. This initiative will explore potential risk factors and biomarkers for colorectal cancer across different patient age groups.

Generally, younger patients tend to be more dismissive with clinical symptoms, particularly those that are nonspecific. Common ones associated with CRC, such as reduced appetite, abdominal pain and weight loss, are frequently overlooked [49]. This leads to a delay in diagnosis for approximately 6 to 9 months [50]. When colorectal cancer presents with symptoms, it often indicates a more advanced stage of the disease, which is linked to a worse outcome. A multicenter retrospective analysis revealed that over 61% of EOCRC patients were diagnosed with metastatic disease, in contrast to 44% of LOCRC patients [51,52]. Due to the general clinical condition, younger patients tend to better tolerate aggressive chemotherapy regimens but, actually, age plays a significant role in prognosis, with younger individuals generally having a more unfavorable outcome [53]. Lastly, in terms of treatment, there are no major differences between EOCRC and LOCRC, with the observation that patients below 50 years are more likely to receive and better tolerate postoperative treatment [54]. Hoping to provide assistance, an international multidisciplinary team (DIRECt) compiled evidence-based guidelines to support clinicians in treating patients with EOCRC [55].

The current recommendations advise beginning screening at 45 years of age for the general population, although modest benefits have been observed in individuals between 45 and 49 years old, compared to the 50 to 75 years age group [56]. Popular methods for screening encompass colonoscopy, flexible sigmoidoscopy, CT colonography, the fecal immunochemical test (FIT), stool DNA test, and the fecal occult blood test (FOBT); the latter two of which may be used in conjunction with sigmoidoscopy and FIT. Among these, colonoscopy is considered the gold-standard for CRC screening [57]. The ultimate goal is to tailor colorectal cancer screening based on an individual’s family history. Under this guidance, individuals with a first-degree relative diagnosed with colorectal cancer or an advanced adenoma before age 60, or with two first-degree relatives diagnosed at any age, should start screening with a colonoscopy at age 40 or ten years younger than the earliest diagnosis in their family [58]. Unfortunately, most EOCRCs are still sporadic, and a significant number of patients who have a critical inherited risk factor are not well-informed about their family medical history, frequently due to inadequate communication among family members. Subsequently, this vital information is often not obtained by physicians during consultations.

Another particular aspect of the case we reported is the use of **conversion therapy**. As previously stated, due to the absence of distinct symptoms, numerous patients are diagnosed with metastatic disease. However, it is essential to determine which patients have surgically removable metastases, or whose disease, initially deemed inoperable, could become operable after obtaining response from conversion systemic therapy. Patients who undergo surgery have median survival rates that are two to three times higher than those treated solely with systemic therapy, thereby providing an opportunity for a cure [59].

When it comes to assessing the efficacy of different systemic therapies, the overall response rate (ORR) appears to be the optimal parameter for both direct comparisons and across different trials. The regimens associated with the highest response rates in the CRYSTAL, PRIME and OPUS studies for RAS-wild type tumors involved the use of chemotherapy doublet together with an anti-EGFR agent [60,61,62]. For mutant RAS tumors, 5-FU, leucovorin, oxaliplatin, irinotecan (FOLFOXIRI) + Bevacizumab would seem to be the best option, as long as the patient’s comorbidities and desired quality of life are taken into account [19,20,21]. In the OLIVIA European phase 2 study, which compared FOLFOXIRI + Bevacizumab to FOLFOX + Bevacizumab, triple chemotherapy had better outcomes (response rates and rate of resections) [21]. However, these results are debatable, because triple therapy is known to increase response rates, and the inclusion of Bevacizumab in both arms makes it impossible to estimate its contribution to the results. Ye et al. used a different approach, with patients being randomized between an arm with and another without cetuximab, both containing FOLFOX/FOLFIRI. Response rates were higher in the cetuximab arm, resulting in a higher rate of resections [63]. Another randomized Asian trial (the BECK study) found that conversion therapy was more effective on left-sided primary tumors than in their right-sided ones (75.0% vs. 30.0%). The 5 year survival for the entire study population was 48.1%. Patients who underwent R0/R1 hepatectomy had a 5 year relapse-free survival (RFS) rate and OS rate of 19.1% and 66.3%, respectively [64]. The majority of the patients who achieved 5 year survival underwent several hepatectomies after local relapses.

Surgical removal of metastases should be scheduled for 3 to 4 weeks after the last dose of chemotherapy alone or combined with anti-EGFR monoclonal antibodies, or at least 5 weeks following chemotherapy combined with Bevacizumab, unless the anti-VEGF agent was excluded from the final cycle. Surgery should be performed as soon as the metastases become technically operable due to a reduction in size, so as to avoid the risk of increased liver toxicity and subsequent higher post-operative complications from excessively prolonged chemotherapy administration [16].

Performing surgery after systemic therapy might present with more difficulties than in cases where patients were initially eligible for resection. Additional local treatments, like portal embolization, combining resection with radiofrequency ablation (RFA) or microwave ablation (MWA), or conducting a two-stage hepatectomy, could be considered. Moreover, patients who do not respond to first-line chemotherapy should still be considered for liver resection, as outcomes following second-line chemotherapy can also be positive. This strategy necessitates regular monitoring to timely determine the optimal moment for surgery [65,66].

Following surgery, resuming the prior systemic treatment regimen may be an option, especially if a significant pathological response was obtained, although randomized studies do not yet provide evidence to back this strategy. Typically, the overall duration of treatment should not surpass six months. Currently, randomized trials are exploring alternative methods, including intraarterial post-operative adjuvant treatments [67].

The effectiveness of conversion therapy in the long term is still debatable. Reports indicate that patients can achieve an overall survival (OS) of around 35 months after undergoing conversion chemotherapy followed by liver resection. However, a significant portion of patients experience early recurrence [68,69]. For properly managing such cases, special commitment and expertise from the multidisciplinary team is demanded, as determining which patients will respond to treatment and become candidates for invasive therapy is less clear, despite great efforts to this extent.

Another important aspect of the case we reported was **the use of liquid biopsy** to guide treatment decisions. Currently, core tumor biopsies represent the gold standard for molecular analyses used in making clinical decisions. However, it is broadly accepted that cancer is a perpetually evolving process, and this temporal and spatial heterogeneity represents a major drawback for the current standard. Against this backdrop, liquid biopsy is emerging as an increasingly prominent, non-invasive option. It offers a complementary, and possibly alternative, approach to overcome these limitations [70,71]. 

Liquid biopsy is a tool that offers the possibility of detecting cancer biomarkers from tumor cells in various bodily fluids, including blood, urine, saliva, feces, and cerebrospinal fluid. The most significant markers analyzed in this approach are circulating tumor DNA (ctDNA) and circulating tumor cells (CTCs), with ctDNA analysis being particularly noteworthy [72]. This way, we could encompass the molecular profile of the disease with numerous advantages, such as its non-invasive nature and the ability to provide rapid results [73].

Colorectal cancer is among the solid tumors that release the largest amounts of ctDNA into the bloodstream [74,75], and the role of liquid biopsy was widely studied in the metastatic setting. It has been established that, in the majority cases, the molecular characteristics of tumors are similar, whether analyzed through ctDNA or direct tissue samples [76,77]. It light of that, several trials concerning practical application in different branches were conducted, such as prognostic information or the guidance of treatment. In cases involving patients with oligometastatic liver disease who underwent surgery with curative intent, the levels of ctDNA could predict a significantly reduced risk of recurrence if a continual decrease to near disappearance were to be observed post-surgery, as opposed to scenarios where levels either remained stable or increased [78,79]. It is noteworthy that ctDNA was successfully cleared in two-thirds of patients who underwent post-operative treatment, highlighting the efficacy of such treatments in this context [80].

Moreover, ctDNA was also studied for its ability to dynamically observe the molecular progression of nonresectable metastatic CRC through different treatment phases. The oscillating patterns of tumor-specific mutations in ctDNA over time provided scientific justification for rechallenging anti-EGFR therapies, which were initially chosen based on clinical empiricism [81]. Two retrospective studies found that patients with mCRC who had no mutations in RAS genes experienced a notably better response and longer progression-free survival when treated again with anti-EGFR agents, in contrast to patients with RAS mutations in ctDNA [29,82]. Recently, the CHRONOS trial provided prospective evidence supporting this theory [83]. In CHRONOS, the subjects who were undergoing their third or subsequent line of treatment underwent ctDNA assessments for RAS, BRAF, and EGFR ectodomain mutations. Patients were rechallenged with anti-EGFR therapy only if they tested negative. This approach led to a 30% response rate and a 63% disease control rate, numbers that compare favorably to those from anti-EGFR rechallenge trials that selected patients based on empirical judgment, as well as to the outcomes from standard chemotherapy treatments for advanced-stage mCRC [83,84,85]. Therefore, the rechallenge with anti-EGFR treatments is a real-life clinical context that is expected to be among the first to benefit from the implementation of interventional ctDNA testing.

CtDNA in colorectal cancer has been initially used to assess tumor burden (detect minimal residual disease) or to investigate molecular changes that can predict treatment response. Nonetheless, advances in our knowledge of the cancer genome, together with the growing accessibility and decreasing costs of sequencing technologies, are opening avenues for analyzing novel biomarkers in ctDNA [86,87]. Currently, there is an ongoing debate about the significance of Tumor Mutational Burden (TMB) in CRC and other solid tumors, especially concerning its connection to the efficacy of immunotherapy. This interest has been amplified by the Food and Drug Administration’s (FDA) recent endorsement of TMB as a comprehensive biomarker for guiding the use of drugs such as Pembrolizumab or Dostarlimab in cancer treatment [88,89]. TMB refers to the quantity of mutations within a megabase of DNA (expressed as Mut/Mb) and, in the context of colorectal cancer, it tends to be highly correlated with microsatellite instability or detrimental mutations in the proofreading segments of the DNA polymerases POLE and POLD. This elevation leads to an increased generation of tumor neoantigens, potentially improving the response to agents that target immune checkpoints [90]. Currently, tumor tissue samples are the gold standard for assessing TMB, even though the inherent variability within a tumor presents a significant challenge to accurately measure it [91]. The ARETHUSA trial is exploring the use of ctDNA to measure TMB as a way to predict how well patients with O-6-Methylguanine-DNA Methyltransferase (MGMT) methylated mCRC respond to immunotherapy following initial treatment with Temozolomide. While this biomarker holds potential, the specific chromosomal regions for its calculation, the minimum number of mutations needed, and the threshold values are yet to be standardized [92]. This is despite widespread international efforts to unify how TMB is assessed and reported.

At the moment, microsatellite instability is the primary biomarker for predicting responsiveness to immunotherapy in mCRC, typically measured in solid tissue samples. Nevertheless, like TMB, the MSI status is affected by spatial and temporal variability [93], which makes tracking it through liquid biopsy valuable for therapeutic purposes.

Another particularity of the case we reported was the presence of **acrometastasis**. 

Metastatic bone disease represents the most prevalent form of bone malignancy. However, metastases located beyond the elbow and knee regions, known as acrometastases, are uncommon, comprising around 0.1% of all instances [94]. Generally, distal bones contain less red bone marrow, making them less commonly involved in diseases. In up to 10% of cases, acrometastases may serve as the initial indication of an underlying cancer [95]. 

The literature commonly identifies lung tumors as the predominant source of acrometastases [94,95,96,97]. Yet, in certain studies focusing on the lower limbs, kidney tumors emerge as the most frequent culprits [95,98]. Among these cases, the tibia is the bone most commonly impacted by acrometastasis [94,99]. Once the acrometastases occur, it is often a signal of a poor prognosis, as they typically present in patients with widespread disseminated disease [100].

The predominant clinical manifestations of acrometastases include pain, a palpable mass, or a mechanical dysfunction in the limb that hinders daily activities. The pain associated with these conditions is often described as deep, intermittent, and not alleviated by standard pain relief methods. Initially, the lesion might not be sensitive to touch, but as it advances, it can mimic an inflammatory condition characterized by swelling, redness, ulceration, or bleeding sites. Differential diagnosis for these symptoms should consider osteomyelitis, rheumatoid arthritis, tenosynovitis and gout [94,101,102,103].

Radiographic imaging often reveals a permeative lesion with destructive characteristics [104]. Lung and renal cell carcinomas usually present as solitary, lytic lesions affecting a single bone in the hand, whereas metastases from breast cancer can appear sclerotic, lytic, or of mixed type and frequently involve multiple lesions [104]. CT scans of the hand and foot are generally of limited utility due to their insufficient resolution in these confined areas. MRI, however, is beneficial for assessing disease within the bone marrow and the tumor’s extension beyond the bone [105].

To obtain a tissue diagnosis, fine needle aspiration (FNA) cytology or trocar biopsy are recommended methods, while incisional biopsy should be avoided [105,106]. Following the determination of the disease stage, treatment options may include surgical intervention through amputation or limb-sparing and reconstructive surgery, as well as external beam radiation therapy and chemotherapy.

The small prevalence of acrometastases reflects in the lack of standardized treatment protocols. Given the poor prognosis associated with these patients, treatment primarily focuses on palliative care. Objectives include alleviating pain, achieving satisfactory tumor removal, ensuring rapid recovery, and maintaining quality of life through the maximal preservation of limb function [101,107]. Typically, administering a single fraction of 8 Gy radiation effectively mitigates symptoms [108]. Bisphosphonates, by inhibiting osteoclast activity, can reduce bone resorption, alleviate pain, lessen skeletal complications, and potentially prolong survival [109]. For tumors that show minimal response to radiation therapy and chemotherapy, a more aggressive surgical strategy is advised [105].

Generally, patients with acrometastases cancer have a poor prognosis, with the average survival time post-diagnosis being under 6 months [108]. Unfortunately, no statistically significant variation in survival rates has been noted based on the location and number, the patients’ age or the histological type of the primary cancer.

## 4. Conclusions

Metastatic rectal cancer requires a multidisciplinary and individualized approach. Despite the typically poor prognosis of mCRC, a subset of patients achieves remarkable long-term survival. Understanding the factors contributing to this phenomenon can guide clinicians to identify patients who will benefit from aggressive treatment strategies. 

We reported the case of a young women diagnosed with metastatic rectal cancer. An individualized therapeutic plan with a multidisciplinary approach was implemented, and a remarkable five-year survival rate was achieved. This case is particular not only because of the multimodal approach, but also because of the age of the patient and it highlights the unfortunate rise of young-onset colorectal cancer.

A particular situation is the management of patients who are potential candidates for conversion therapy, a method deemed to transform cases initially regarded as unresectable to candidates for surgical intervention, via systemic therapy. It is important to approach each case individually, as every patient with limited liver disease should be considered as a candidate for secondary resection. Moreover, liquid biopsy has an important role in the individualized management of mCRC patients, as it offers additional information for treatment decisions.

## Figures and Tables

**Figure 1 medicina-60-00696-f001:**
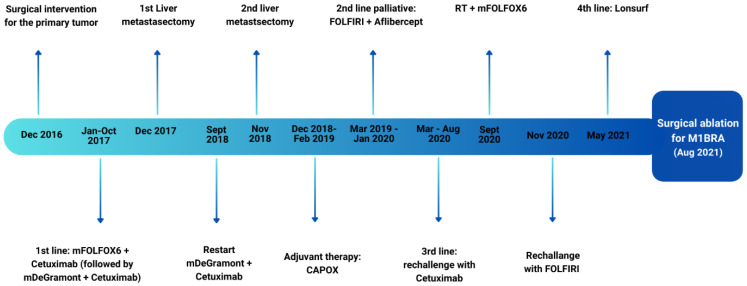
Timeline and overview of the multimodal treatment received by the patient.

**Figure 2 medicina-60-00696-f002:**
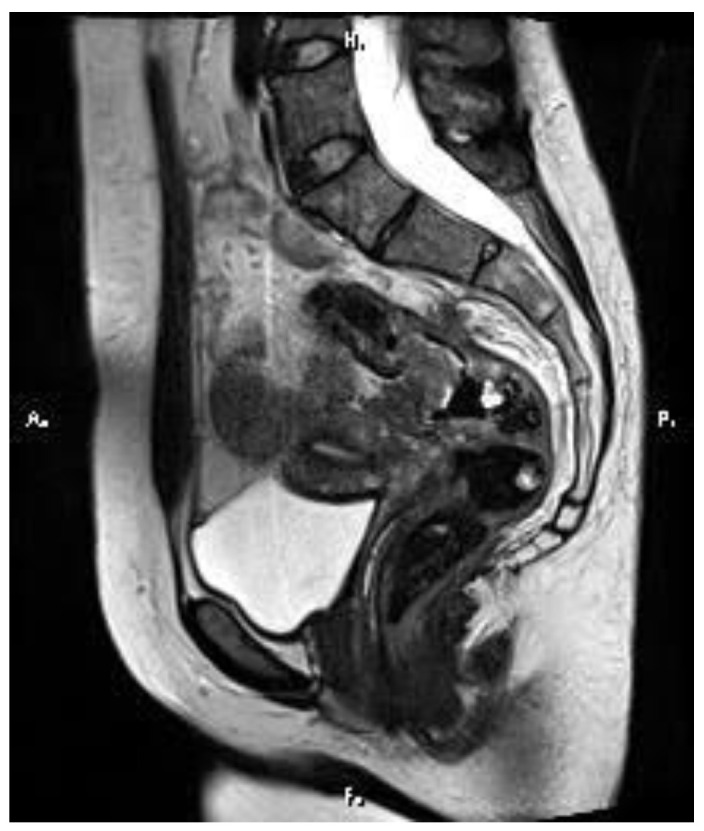
Baseline MRI showing an infiltrative rectal tumor which extends over a length of 8.5 cm, stenotic, that exceeds the serosa and infiltrates the perirectal space, the postero-superior part of the uterus, and the peritoneum of the pouch of the Douglas.

**Figure 3 medicina-60-00696-f003:**
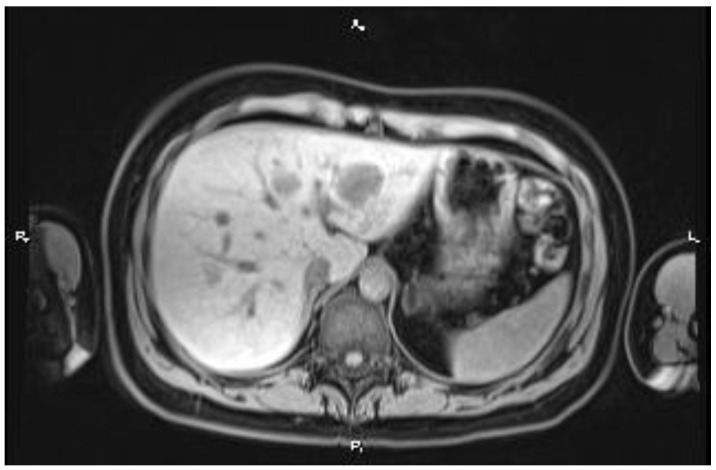
Baseline MRI showing multiple liver metastases, relatively well delimited, with sizes between 2–3 mm and 3 cm, distributed diffusely in all segments of liver parenchyma.

**Figure 4 medicina-60-00696-f004:**
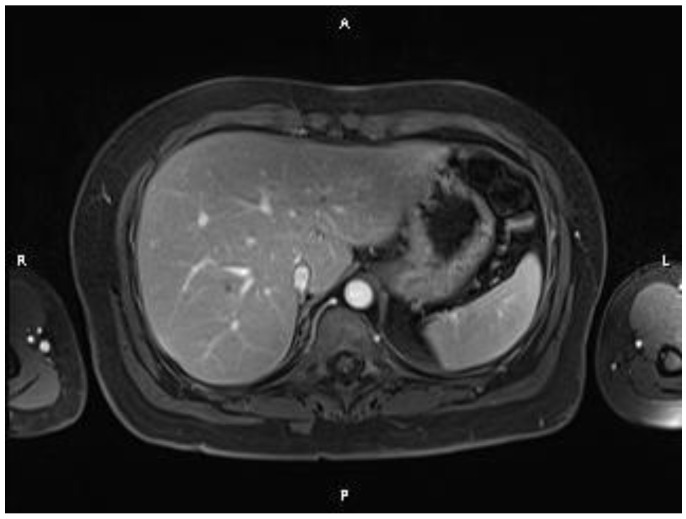
MRI showing a decrease in size and number of liver metastases (partial response).

**Figure 5 medicina-60-00696-f005:**
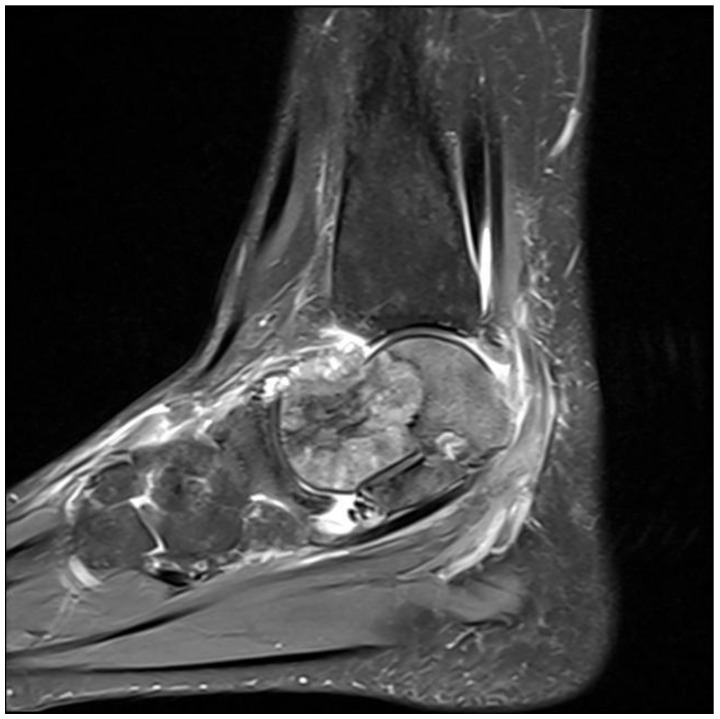
MRI showing a metastasis in the left talar bone measuring 33/31/23 mm.

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
