# Peer review of "Multimodal Treatment of Metastatic Rectal Cancer in a Young Patient: Case Report and Literature Review"

_medicina, 2024, doi:10.3390/medicina60050696_

Round 1
Reviewer 1 Report
Comments and Suggestions for Authors
Tlili Nizar
Response to Editor
Journal: Medicina (ISSN 1648-9144)
Article Type: Review
Manuscript ID: medicina-2923152
Title: The Importance of Multidisciplinary Approach and the Advantages of Liquid Biopsy in Metastatic Rectal Cancer: A Case Report and Literature Review
This article will focus on the case of a young woman who was diagnosed with metastatic rectal cancer, but I think that in a Review Article it is necessary to collect all the information that relates to the subject and not present results for just one clinical case. Furthemore, the research idea lacks innovation and the workload seems to be insufficient. It appears that this study is not appropriate for publishing in the“medicina“.
Other comments:
- The title is poorly presented: it must be changed to highlight the work.
- The abstract is too short and does not reflect the objective of this study.
- The abbreviations must be put in the first place of the article.
-Introduction: the introdcution is too summary, it must be enriched with recent references- Abstract: the objective of this review is not reported.
- The general structure of the manuscript is not acceptable, for example, the figures are blurry and lack symbol and legends.
- The references are poorly written and are not updated and not standardized, some journal in italic et complete name and others with abreviation.
-Comments on the Quality of English Language: Extensive editing of English language is required.

Comments on the Quality of English LanguageExtensive editing of English language is required.
Author Response
This article will focus on the case of a young woman who was diagnosed with metastatic
rectal cancer, but I think that in a Review Article it is necessary to collect all the
information that relates to the subject and not present results for just one clinical case.
Furthemore, the research idea lacks innovation and the workload seems to be
insufficient. It appears that this study is not appropriate for publishing in the “medicina“.
Other comments:
- The title is poorly presented: it must be changed to highlight the work.
Thank you for your constructive comments. We have tried to reorganize our article according to your suggestions and comments.
We have extended the article and the review of the literature focused on the
particularities of the case we reported: disease's early onset, the use of conversion therapy, the application of liquid biopsy to guide treatment, and the specific nature
of the bone metastasis
The title was changed to: “Multimodal Treatment of Metastatic Rectal Cancer in a
Young Patient: Case Report and Literature Review”
- The abstract is too short and does not reflect the objective of this study.
We have changed the abstract to be more comprehensive. The aim of the study is
stated in the abstract.
- The abbreviations must be put in the first place of the article.
The abbreviations were written out in full on first use.
-Introduction: the introduction is too summary, it must be enriched with recent
references- Abstract: the objective of this review is not reported.
We have extended the introduction according to your suggestion.
- The general structure of the manuscript is not acceptable, for example, the figures are
blurry and lack symbol and legends.
We have replaced the figures with others that have better quality.
- The references are poorly written and are not updated and not standardized, some
journal in italic et complete name and others with abreviation.
The references were inserted in the text with EndNote.
-Comments on the Quality of English Language: Extensive editing of English language is
required.
We have reviewed the quality of English language in the article.

Reviewer 2 Report
Comments and Suggestions for Authors
This article is fairly straightforward and, to the best of my knowledge, in line with current trends in metastatic colorectal cancer therapy.
I'd just advise to classify it as a case report rather than a review, as this seems to be the main focus of the current study.
Author Response
This article is fairly straightforward and, to the best of my knowledge, in line with current trends in metastatic colorectal cancer therapy. I'd just advise to classify it as a case report rather than a review, as this seems to be the main focus of the current study.
Thank you for your comments. We have changed the structure of the article and extended the review of the literature with emphasis on the particularities of the case we reported: disease's early onset, the use of conversion therapy, the application of liquid biopsy to guide treatment, and the specific nature of the bone metastasis.

Reviewer 3 Report
Comments and Suggestions for Authors
Dear authors,
thank you for analysing and discussing this exemplary case of a young metastatic cancer patient in detail.
I have a few comments and I hope you can address them:
1) Can you add a short paragraph stating whether you agree that in rectal cancer there is a trend for a higher incidence of advanced cancer in younger patients? If you agree, may you plese add a few hypotheses to explain this phenomenon.
2) Have you done molecular pathology on the liver metastasis and compared the results with the original tumour? Have you performed any kind of genetic comparison between the initial tumour and the resected metastasis (liver, brain)?
3) You wrote that "a remarkable five-year survival was achieved". Overall survival is an important endpoint in rectal cancer, but can you add some information about quality of life? Has this young woman been able to carry on with her life (personal, professional)? I would appreciate it if you could add some information about the impact of the treatment and disease progression on her quality of life to better understand the price she may had to pay for prolonged survival.
Thanks
Author Response
1) Can you add a short paragraph stating whether you agree that in rectal cancer there is a trend for a higher incidence of advanced cancer in younger patients? If you agree, may you plese add a few hypotheses to explain this phenomenon.
Thank you for your comments. We have added more data in the discussions regarding early onset colorectal cancer.
2) Have you done molecular pathology on the liver metastasis and compared the results with the original tumour? Have you performed any kind of genetic comparison between the initial tumour and the resected metastasis (liver, brain)?
Thank you for your comment. We have included information in the case report that helps clarify this specific issue.
3) You wrote that "a remarkable five-year survival was achieved". Overall survival is an important endpoint in rectal cancer, but can you add some information about quality of life? Has this young woman been able to carry on with her life (personal, professional)? I would appreciate it if you could add some information about the impact of the treatment and disease progression on her quality of life to better understand the price she may had to pay for prolonged survival.
Thank you for your comment. We have included in the case report information about the patient’s quality of life along the development of the disease.

Round 2
Reviewer 1 Report
Comments and Suggestions for Authors
No comments
Comments on the Quality of English LanguageMinor editing of English language required